# Assessment of endophytic bacterial diversity in rose by high-throughput sequencing analysis

**Ao-Nan Xia[1,2], Jun Liu[1], Da-Cheng Kang[2], Hai-Guang Zhang[2], Ru-Hua Zhang[2], Yun-Guo Liu📧[2]***

**1** College of Life Science and Technology, Xinjiang University, Urumqi, China, **2** College of Life Sciences, Linyi University, Linyi, China

* yguoliu@163.com

**Data Availability Statement:** All relevant data are within the manuscript.

## Abstract

The endophytic bacterial diversity of rose was analyzed by high-throughput sequencing of 16S rDNA and functional prediction of the bacterial community. The number of bacterial sequence reads obtained from 18 rose samples ranged from 63,951 to 114,833, and reads were allocated to 1982 OTUs based on sequences of the V3-V4 region. The highest Shannon Index was found in Luogang rose (1.93), while the lowest was found in Grasse rose (0.35). The bacterial sequence reads were grouped into three different phyla: Firmicutes, Proteobacteria, and Actinobacteria. At the genus level, *Bacillus* and *Staphylococcus* had the highest abundance across all 18 samples; *Bacillus* was particularly abundant in Daguo rose (99.09%), *Rosa damascena* (99.65%), and Fenghua rose (99.58%). Unclassified OTUs were also found in all samples. PICRUSt gene prediction revealed that each endophyte sample contained multiple KEGG functional modules related to human metabolism and health. A high abundance of functional genes were involved in (1) Amino Acid Metabolism, (2) Carbohydrate Metabolism, (3) Cellular Processes and Signaling, (4) Energy Metabolism, and (5) Membrane Transport, indicating that the endophytic community comprised a wide variety of microorganisms and genes that could be used for further studies. The rose endophytic bacterial community is rich in diversity; community composition varies among roses and contains functional information related to human health.

## Introduction

Rose is a deciduous shrub of the genus *Rosa* from the Rosaceae family. Roses typically grow in sunny locations and are tolerant of cold and drought. They are perennial flowering plants found throughout the world, especially in subtropical and temperate regions of the northern hemisphere [1]. The species and community composition of endophytic bacteria are essential to the growth of roses, affecting both their quality and their susceptibility to disease. Endophytic bacteria are a group of microorganisms that are symbiotic, parasitic, or saprophytic on host plants [2]. They can colonize niches similar to vascular wilt endophytes from *Artemisia*

**Funding:** This work was supported by grant from the High-level talent introduction project of Linyi University (LYDX2018BS032).

**Competing interests:** The authors have declared that no competing interests exist.

*nilagirica* (Clarke) Pamp, exhibit antibacterial properties against human pathogens, and produce enzymes of multidrug resistance similar to clinical strains [3]. Some substances secreted from bacteria affect plant physiology by interacting with plant growth regulators. For example, interactions between endophytes and *Echinacea* affect plant secondary metabolite content, bacterial colonization specificity, and plant growth [4]. Endophytic bacterial diversity is also an important resource for the treatment of environmental pollutants and the improvement of human health. Islam et al. (2019) found that endophytic bacteria from *Ginkgo biloba* were potential candidates for controlling serious foodborne pathogens, either by themselves or through their metabolites [5]. The physiological functions of endophytic bacteria have also received increased attention. Hung et al. (2007) found that most of the isolates from soybean were motile and produced indole acetic acid; 70% and 33% of the isolates secreted cellulase and pectinase, respectively [6]. Pan et al. (2015) isolated four endophytic bacterial strains from wheat, all of which significantly reduced fungal growth and spore germination of *Fusarium graminearum* [7]. Microorganisms may benefit from the supply of nutrients to plant pathogens, making them beneficial as potential biocontrol agents against blight [8]. On the other hand, plants may increase the absorption of nutrients due to the presence of microorganisms in their tissues, enhancing their ability to survive adversity [9].

Culture-based and non-culture-based techniques were used to study the microbial diversity of endophytic bacteria. Although often using a culture-based approach, they cannot capture non-culturable microorganisms. Therefore, molecular biology-based methods are more accurate and reliable for identifying culturable and non-cultivable microorganisms [10]. High-throughput sequencing produces large amounts of data and shows good repeatability between samples. It greatly expands the field of microbiology ecology, more accurately identifies microbial diversity and flora structure, including those that are difficult to cultivate and / or exist at low levels, and enables a more comprehensive analysis of microbial diversity [11]. PICRUS is a technique that analyzes the functional composition of an existing sequenced microbial genome, and infers the composition of functional genes in the sample through 16S data and a reference genome database to analyze the functional differences between different samples and groups. and is widely used in humans, soils, plants and other mammals. The predicted correlation between the gene content and the metagenomic assay was 84%-95%, and the functional analysis of intestinal microbial flora and soil flora was close to 95%, which could well reflect the functional gene composition in the samples.[12].

Rose is an important economic and medicinal crop. At present, research on roses focuses mainly on flower quality evaluation, processing techniques, and the extraction of pigments and essential oils [13–15]. Roses are rich in vitamins, amino acids, and many functional components that benefit health conditions such as inflammation, hematemesis, and diarrhea. At the same time, rose possesses cosmetic and skin-moisturizing properties, can help reduce pain, functions as an aromatic deodorant, and invigorates the spleen. Its medicinal value is extremely high [16–18]. It is therefore worthwhile to explore the diversity and function of rose endophytic bacteria because they affect the functional properties of the roses themselves. In previous studies, endophytic bacteria of *Glycine max* [6], *Triticum aestivum* [7], *Brassica napus* [19], and *Oryza sativa* [20] were extensively documented; however, endophytic bacteria of roses have received little attention. In present study, high-throughput sequencing and bioinformatic analysis of 16S rDNA was performed to explore the diversity and potential functions of endophytic bacteria from rose. The results of this study will help researchers exploit the beneficial endophyte resources of rose, screen for beneficial microorganisms in rose petals and control potentially harmful endophytes. Provide a certain reference for some functional gene research and screening of specific gene functional microorganisms through PICRUSt function analysis.

## Materials and methods

### Sample preparation

The 18 rose varieties have a large planting area and a wide planting area in China. Each rose is planted over 10hm$^2$ at the sampling site. Eighteen rose varieties were selected, including *Rosa rugosa* x *Rosa sertata* (KS, Gansu), Hetian rose (HT, Xinjiang), *Rosa rugosa* 'Alba' (BJ, Beijing), Luogang rose (GL, Guangzhou), Gansu *Rosa rugosa* x *Rosa sertata* (GKS, Gansu), Grasse rose (GLS), *Rosa davurica* x *Rosa rugosa* 'Plena' (Zr), Fenghua rose (FH), Daguo rose (DG), Heze rose (HZ), *Rosa damascena* (DM), *Rosa rugosa* 'Plena' (CBH), Shanci rose (PY), *Rosa rugosa* 'Alba Plena' (CBB), *Bulgaria red rose* (BH), *Soviet rose* (SL), *Bulgaria white rose* (BB), and *Rose centifolia* (XS) from Shandong province. The rose samples were delivered to the laboratory at low temperature (4˚C).

Rose samples were surface sterilized according to the protocol of [21] with some modifications. The surface of the rose sample is sterilized by rinsing with sterile water, and the last rinsed water (0.01mL) were spread onto the Plate Count Agar (PCA) to check the sterilization effect. Samples with completely sterile surfaces were rinsed with distilled water more than three times to remove the remaining microbial DNA. Approximately 2.0 g of surface-sterilized sample was ground in a mortar with quartz sand, then placed in 9 ml of normal saline. Samples were serially diluted, spread on Luria-Bertani (LB) liquid media plates, and incubated at 37˚C for 48–72 h. All microorganisms were preserved at –20˚C in LB broth containing 10% glycerol (v/v) and freeze-dried.

### DNA extraction and Illumina high-throughput sequencing

Microbial DNA was extracted from the samples using the E.Z.N.A.$^{TM}$ Mag-Bind Soil DNA Kit (OMEGA, USA), and the concentration of DNA in each extraction was determined using 1% agarose gel electrophoresis.

The V3 and V4 variable regions of the bacterial 16S rDNA gene were amplified using the primer 341F (`CCTACGGGNGGCWGCAG`) and 805R (`GACTACHVGGGTATCTAATCC`). The DNA was denatured with the following protocol: 95˚C for 3 min, followed by 27 cycles of 95˚C for 45 s, 55˚C for 30 s, and 72˚C for 45 s, with a final extension of 72˚C for 10 min. The PCR products were purified and then sequenced using MiSeq Illumina platform (Illumina, USA) at Sangon Biotech Co, Ltd (Shanghai, China) [22, 23].

### Data analysis

High quality sequences are extracted using CASAVA packages. The raw sequences were filtered by length and quality, amplicon primers were removed, and small fragments were thrown away. The unique sequence set was classified into operational taxonomic units (OTUs) with UC LUST, using a similarity threshold of 97% identity. The R Venn Diagram package (1.6.16) was used to analyze the numbers of shared and unique OTUs among the 18 samples [24]. Chao1 and ACE indexes were used to estimate the OTU richness, and Shannon and Simpson indexes were used to evaluate the bacterial diversity. Higher Shannon index and lower Simpson index indicate higher microbial diversity [25, 26]. The principal component analysis (PCA) method is designed to use the idea of dimensionality reduction to transform multiple indicators into a few comprehensive indicators, which can be used to analyze the microbial community composition of samples [27]. Heat map analysis and a Bray-Curtis-based multiple sample similarity tree were used to examine similarities and differences in microbial community structure among the samples. We used the PICRUSt package to infer the potential genetic capacity of bacterial communities, to assess the specific contribution of

individual taxa to the metagenic genome, and to annotate the predictive functional genes using the KEGG database [12].

## Results

### Richness and diversity analysis of bacterial communities

To investigate microbial community composition, reads were classified to OTUs to identify bacterial microorganisms present in rose samples. The final number of reads of the bacterial sequence in each sample ranged from 63,951 to 114,833. The reads were allocated to 1982 OTUs based on a similarity threshold of 97% identity.

### Rarefaction analysis

Rarefaction curve can be used to compare the richness of samples with different sequencing Numbers, and to evaluate whether the sample size represents the diversity of the original samples [28]. The rarefaction and Shannon curves of bacterial communities classified based on 97% similarity OTU are shown in Fig 1A and 1B. Bacterial diversity reached an asymptote, which indicated that this sequence could well represent the bacterial diversity of 18 rose samples.

### Bacterial alpha diversity

Bacterial alpha diversity indices, including the Chao1, ACE, Shannon, and Simpson indices, are presented in Table 1. In all samples, the Good's coverage of bacterial OTUs was 100%, indicating that the major bacterial OTUs had been captured [29]. The CBH sample had the highest values for the Chao1 and ACE indices, indicating that its richness was high compared to the other samples. The GL sample had the highest value of the Shannon index and the lowest value of the Simpson index. It shows that the diversity is higher than other samples.

### Diversity among bacterial communities in rose samples

A Venn diagram was established to evaluate the distribution of OTUs among different samples (Fig 2A). Between 5 and 50 OTUs were obtained from the 18 rose samples. The GL sample had the greatest number (50 OTUs), and the DG sample had the fewest (5 OTUs). Only one OTU was shared among all 18 rose samples, highlighting the fact that the bacterial communities of different roses are very different.

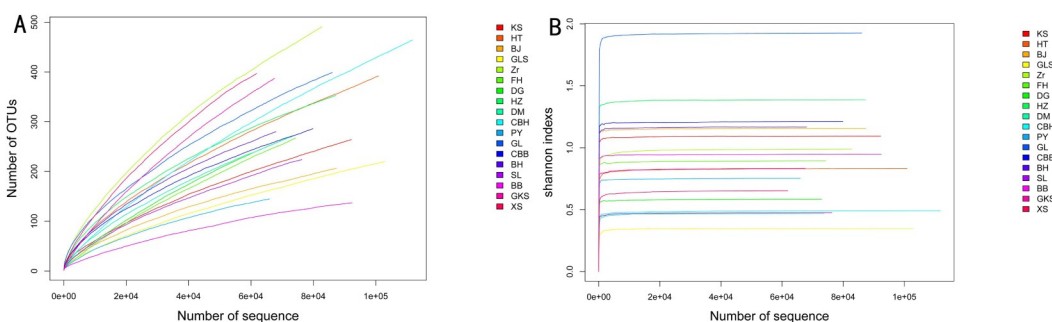

**Fig 1.** Rarefaction curves (A) and Shannon curves (B) of endophytic bacteria of rose with 97% similar of 16S rDNA.

**Table 1. The Operational Taxonomic Units (OTUs) for rose.**

| Sample | Seq-num | OTU-num | Shannon-index | ACE-index | Chao1-index | Coverage | Simpson |
|---|---|---|---|---|---|---|---|
| BB | 92459 | 137 | 0.95 | 383.27 | 208.08 | 1.00 | 0.50 |
| BH | 68010 | 280 | 1.17 | 968.73 | 567.02 | 1.00 | 0.50 |
| BJ | 87420 | 206 | 1.15 | 720.26 | 378.86 | 1.00 | 0.46 |
| CBB | 79877 | 286 | 1.21 | 990.05 | 597.25 | 1.00 | 0.51 |
| CBH | 111761 | 464 | 0.49 | 2460.09 | 1131.81 | 1.00 | 0.79 |
| DG | 72940 | 271 | 0.59 | 1578.86 | 905.41 | 1.00 | 0.76 |
| DM | 73774 | 272 | 0.47 | 1947.12 | 884.62 | 1.00 | 0.84 |
| FH | 74324 | 267 | 0.89 | 878.92 | 795.95 | 1.00 | 0.51 |
| GKS | 67582 | 387 | 0.83 | 1430.72 | 867.48 | 1.00 | 0.59 |
| GL | 86065 | 399 | 1.93 | 1355.51 | 720.47 | 1.00 | 0.22 |
| GLS | 102820 | 220 | 0.35 | 1137.62 | 597.50 | 1.00 | 0.89 |
| HT | 100896 | 392 | 0.83 | 2077.18 | 977.67 | 1.00 | 0.65 |
| HZ | 87272 | 353 | 1.39 | 999.49 | 630.82 | 1.00 | 0.32 |
| KS | 92229 | 264 | 1.09 | 1336.59 | 618.90 | 1.00 | 0.53 |
| PY | 65964 | 145 | 0.75 | 408.09 | 268.00 | 1.00 | 0.54 |
| SL | 76317 | 224 | 0.48 | 1131.93 | 554.78 | 1.00 | 0.81 |
| XS | 61837 | 397 | 0.65 | 1237.03 | 730.49 | 1.00 | 0.75 |
| Zr | 82760 | 491 | 0.99 | 1089.63 | 1101.50 | 1.00 | 0.58 |

## Relative abundance of bacterial communities

A total of three phyla, six classes, 13 orders, 24 families, and 38 genera were identified by Ribosomal Database Project (RDP) classifier. Fig 3A and 3B show the relative abundance of bacteria at the phylum and genus levels, respectively. Only phyla with a relative abundance values ≥ 0.01% of the bacterial community are displayed. The bacterial sequence reads were grouped into three different phyla: Firmicutes, Proteobacteria, and Actinobacteria. Firmicutes had the highest total abundance in all samples, especially in HZ (100%) and Zr (100%). At the genus level, the top five bacterial groups were *Bacillus*, *Staphylococcus*, *Pantoea*, *Paenibacillus*, and unclassified. These five groups showed significant differences in composition among samples. *Bacillus* had the highest total abundance in all 18 samples, particularly in DG (99.09%), DM (99.65%), and FH (99.58%). *Pediococcus* was detected only in the BB sample (65.57%); in

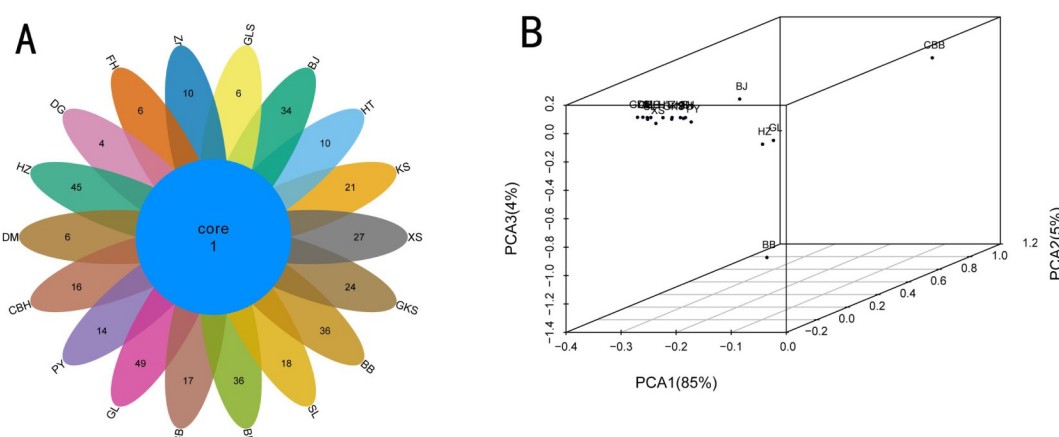

**Fig 2.** Venn diagrams (A) and PCA analysis (B) of the multiple samples according to bacterial diversity.

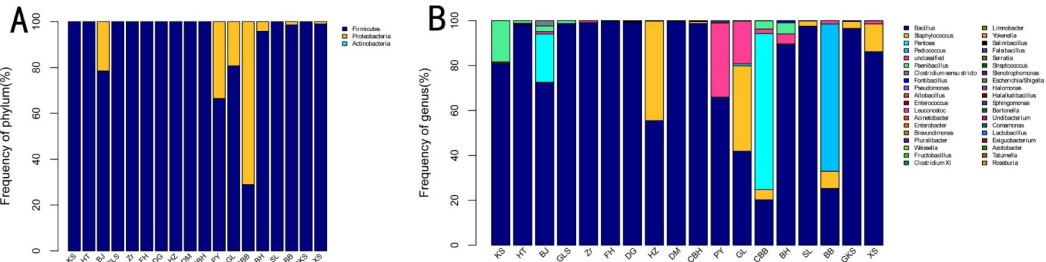

**Fig 3.** Relative abundance of bacterial community at phylum (A) and genus (B) level based on 97% sequence similarity.

the other samples, its abundance was extremely low (<0.01%) or undetectable. Sequences that were designated as unclassified could not be classified based on currently available taxonomic reference data. An unclassified genus was the predominant genus in the GL (19.04%) and PY samples (33.02%), indicating that a large number of unknown genera may exist in some rose samples.

The bacterial community PCA is shown Fig 2B. There was significant separation of bacterial communities from different samples, and the first, second, and third PC axes explained 85%, 5% and 4% of the variance in bacterial species, respectively. The analysis revealed similarity between the GL and HZ samples and among the HT, KS, KS, XS, GKS, SL, BH, PY, CBH, DM, DG, FH, Zr, and GLS samples. These results confirmed that bacterial species composition among the 18 rose samples was extremely variable.

## Bacterial community comparisons

Fig 4A shows a heat map of bacterial abundance at the genus level for the 18 endophytic bacterial communities. The heat map can reflect the relative abundance of the bacterial community by color changes, and the blue to red gradient indicates the relative abundance from low to high. The heat map also shows the high abundance of *Bacillus* in all samples, as well as the

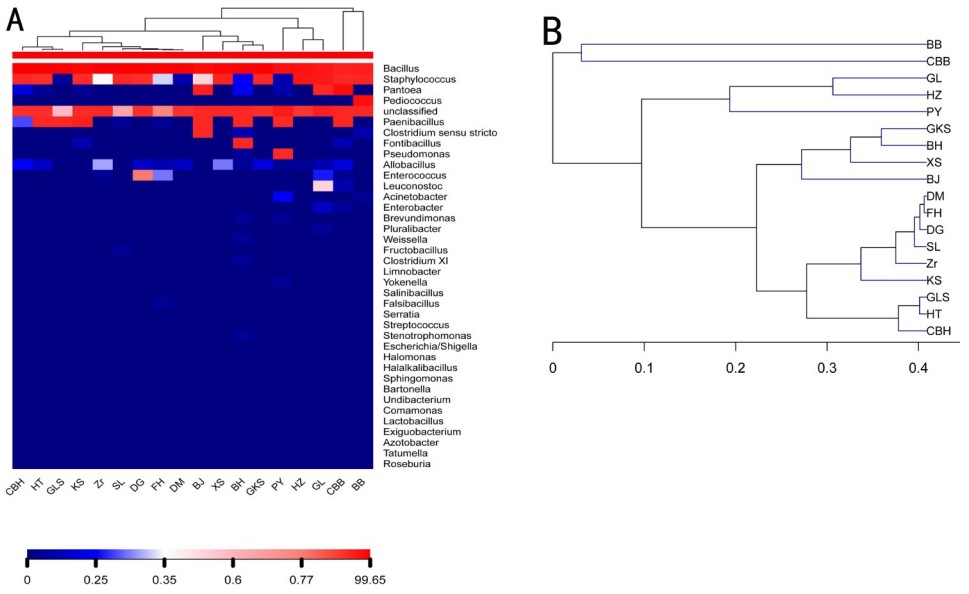

**Fig 4.** Bacterial community heat map analysis at the genus level (A). Hierarchical cluster tree for bacteria (B).

dominance of *Pediococcus* in BB and its absence in other samples. This result is consistent with the previous analysis of the relative abundance of the bacterial community in Fig 3B.

Bray-curtis dissimilarity was used to evaluate the differences in species composition, and to calculate the quantitative characteristics of species composition in biological samples. Two groups of samples were similar: (1) GL and HZ and (2) HT, KS, KS, XS, GKS, SL, BH, PY, CBH, DM, DG, FH, Zr, and GLS. CBB, BB, and PY existed as unique samples (Fig 4B). This result is consistent with the previous analysis of the relative abundance analysis for the bacterial communities.

## Functional gene prediction

The heat map results based on PICRUSt functional gene predictive analysis is presented in Fig 5. The heat map can reflect the relative abundance of gene function predicted by PICRUSt through color changes, and the blue to red gradient indicates the relative abundance from low to high. In this study, we identified 329 different predicted functions, which were divided into 42 functional modules. The most abundant included (1) Amino Acid Metabolism, (2) Carbohydrate Metabolism., (3) Cellular Processes and Signaling, (4) Energy Metabolism, (5) Membrane Transport, (6) Poorly Characterized, (7) Replication and Repair, and (8) Xenobiotic Biodegradation and Metabolism. The heat map indicated that the function abundances of GL, PY and CBB samples were higher than those of other samples.

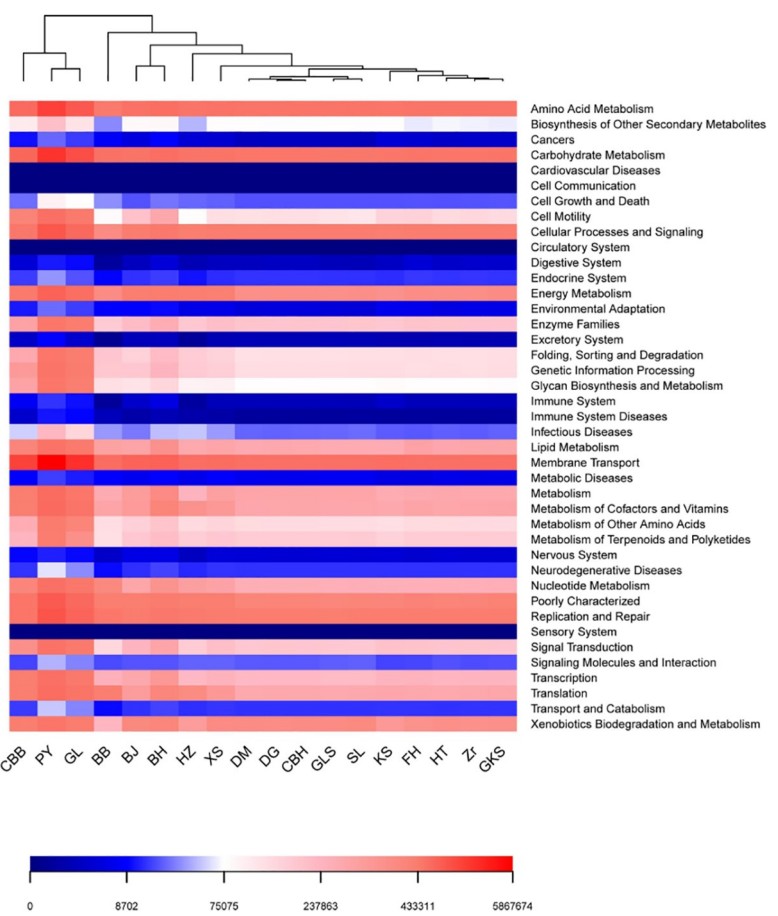

**Fig 5. Heat map of predicted functional pathways assigned to interesting genes investigated in bacteria of rose.**

## Discussion

We assessed the microbial endophyte composition of rose using high-throughput sequencing. Based on the OTUs from each sample, their relative abundances, and Rarefaction analysis (Table 1), indicating that there were abundant endophytic bacteria in the sampled roses. At the phyla level (Fig 3A). These results are consistent with previous findings that endophytic bacteria are mainly distributed among the Firmicutes, Proteobacteria, and Actinobacteria. The most abundant groups in *Trewia nudiflora* were Proteobacteria, Actinobacteria, and Firmicutes [30]. Similarly, Bodenhausen et al. (2013) analyzed the microbial diversity of *Arabidopsis thaliana* leaves and roots by 454 pyrophosphate, and the dominant bacteria were proteobacteria, actinobacteria and Bacteroidetes [31], and Proteobacteria, Firmicutes, Bacteroides, and Actinobacteria were the dominant phyla in *Brassica napus* seeds [32]. At the phyla level (Fig 3B), *Bacillus* may therefore be a core genus and a common predominant genus in roses. In addition, unclassified OTUs were common in all rose samples; further research is required to identify the unclassified bacteria that constitute such a large percentage of rose bacterial communities. In this study, *Pediococcus* was detected only in BB (65.57%), while in other samples its abundance was extremely low (<0.01%) or undetectable. This result may be due to the cultivar type and/or the growing environment. However, Ozgül et al. (2011) showed that 33 dominant genera were found across the soil, including eight Actinobacteria, six Acidobacteria, five α-Proteobacteria, four γ-Proteobacteria, and three Bacteriodetes [33]. Chen et al. (2018) found that the most abundant groups in *Salvia miltiorrhiza* seeds were γ-Proteobacteria (67.6%), α-Proteobacteria (15.6%), Sphingobacteria (5.0%), and Bacilli (4.6%) [34]. Rybakova et al. (2017) reported that *Ralstonia*, *Acetobacteraceae*, *Bacillus*, and *Mesorhizobium* were the most abundant genera in *Brassica napus* seeds [32]. Our results suggest that the endophytic bacteria of rose are different from other plants at the genus level. This result may reflect differences in soil and soil composition (pH, organic matter, available phosphorus, etc.) that provide different environments for endophytic bacteria. Plant endophytic bacterial communities are also affected by plant type. Both these factors affect microbial diversity and microbial interactions in soils [35]. Beneduzi et al. (2013) analyzed plant growth promotion (PGP) bacteria in sugarcane rhizosphere soil, roots and stems by 16S rDNA PCR-RFLP, and found that soil pH and clay were the factors most closely related to bacterial diversity [36]. Environmental factors in different field locations and years had much more influence on microbial rhizosphere community than plant genotypes. [35]. On the other hand, differences in exudation from different plants led to differences in plant endophyte communities; the planting and growing period of plants have effects on microbial diversity and flora structure under some circumstances. The growth period of plants has an effect on the microbial diversity and microbial community structure of potato Rhizosphere in sandy soil with different organic compounds [37]. Likewise, plant tissue and growth stage had significant effects on the endophytic bacterial community structure of leaves, stems, and roots of *Stellera chamaejasme* in northwestern China [38].

PICRUSt provides a convenient method for predicting the function of endophytic bacteria using functional analysis of metagenomic sequencing data and the 16s prediction function. In this study, PICRUSt was used to predict the gene function of rose petal microbial community based on 16S rRNA amplicon prediction results (Fig 5). Many studies have also emphasized the importance of microbial metabolism and microbial functional genes in cell behavior and activity, natural product biosynthesis, and other metabolic processes. The presence of *H. nitritophilus* and *P. viridiflava* on maize balanced cell osmotic pressure and antagonized the pathogen *Ustilago maydis* [39]. Abia et al. (2017) analyzed river water and sediment microbial diversity through high-throughput analysis and predicted its functional genes through functional maps [40]. Genes relevant to metabolism can influence plant growth, and beneficial

bacteria can synthesize and secrete secondary bio-active metabolites that inhibit the spread of soil disease and maintain host plant health; the endophytic bacteria of rose may perform such functions. A high abundance of functional genes involved in nitrogen metabolism was detected in aging flue-cured tobaccos [41].

## Conclusions

The high-throughput sequencing of 16S rDNA from 18 rose bacterial endophyte communities revealed that the communities were composed of three dominant phyla (*Firmicutes*, *Proteobacteria*, and *Actinobacteria*) and five dominant genera (*Bacillus*, *Staphylococcus*, *Pantoea*, *Paeni-Bacillus*, and *unclassified*). There was no significant difference among rose endophytic bacterial communities at the phylum level, but there were significant differences at the genus level. Gene families related to (1) Amino Acid Metabolism, (2) Carbohydrate Metabolism, (3) Cellular Processes and Signaling, (4) Energy Metabolism, and (5) Membrane Transport were identified in rose endophytic bacteria. The results of this preliminary study show that the endophytic composition of rose has unique characteristics in population composition and potential function. Further research may be able to isolate the rose endophytic bacterial strains and screen them for potential medicinal and industrial applications, such as pest control and the production of antibacterial and anticancer compounds, macromolecule-degrading enzymes, and other active substances.

## Acknowledgments

The author thanks Long-Wei Fu and Yan-Zhen Zhang for their help in processing rose samples and useful discussion.

## Author Contributions

**Conceptualization:** Ao-Nan Xia, Jun Liu, Hai-Guang Zhang, Yun-Guo Liu.

**Data curation:** Ao-Nan Xia.

**Formal analysis:** Ao-Nan Xia.

**Funding acquisition:** Yun-Guo Liu.

**Methodology:** Ao-Nan Xia.

**Resources:** Jun Liu, Yun-Guo Liu.

**Supervision:** Jun Liu, Yun-Guo Liu.

**Writing – original draft:** Ao-Nan Xia.

**Writing – review & editing:** Da-Cheng Kang, Hai-Guang Zhang, Ru-Hua Zhang, Yun-Guo Liu.

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
