## [Decision Letter · Decision Letter 0]

6 Jan 2020

PONE-D-19-30114

Assessment of endophytic bacterial diversity in Rose by high-throughput sequencing analysis

PLOS ONE

Dear Liu_Yun-Guo,

Thank you for submitting your manuscript to PLOS ONE. After careful consideration, we feel that it has merit but does not fully meet PLOS ONE’s publication criteria as it currently stands. Therefore, we invite you to submit a revised version of the manuscript that addresses the points raised during the review process.

We would appreciate receiving your revised manuscript by Feb 20 2020 11:59PM. To enhance the reproducibility of your results, we recommend that if applicable you deposit your laboratory protocols in protocols.io, where a protocol can be assigned its own identifier (DOI) such that it can be cited independently in the future. For instructions see: http://journals.plos.org/plosone/s/submission-guidelines#loc-laboratory-protocols

We look forward to receiving your revised manuscript.

Kind regards,

Aradhana Mishra, Ph.D.

Academic Editor

PLOS ONE

Journal Requirements:

Additional Editor Comments (if provided):

Reviewers' comments:

Reviewer's Responses to Questions

**Comments to the Author**

1. Is the manuscript technically sound, and do the data support the conclusions?

Reviewer #1: Partly

Reviewer #2: No

2. Has the statistical analysis been performed appropriately and rigorously? 

Reviewer #1: I Don't Know

Reviewer #2: No

3. Have the authors made all data underlying the findings in their manuscript fully available?

Reviewer #1: No

Reviewer #2: No

4. Is the manuscript presented in an intelligible fashion and written in standard English?

Reviewer #1: Yes

Reviewer #2: Yes

5. Review Comments to the Author

Reviewer #1: The manuscript reports on assessment of endophytic bacterial diversity in rose using 16 S high throughput sequencing. The paper can be of interest to researchers working in the area of microbial diversity and microbial technologies. However, the paper cannot be accepted in its present form. The authors need to satisfactorily address the following queries:

1. What was the criterion for selecting the 18 rose varieties not mentioned?

2. Whether the sequences were submitted to GenBank is also not mentioned?

3. How many biological and technical replicates were used for sequencing?

4. It would have been better if authors would also have performed gene expression analysis to confirm the abundance of specific sequences as per high throughput sequencing results?

5. Authors should also provide in details a commentary on future perspectives of the present study in the manuscript.

Reviewer #2: Thank you for the opportunity to review this paper, which concerns the composition and potential function of bacterial endophytes of rose. I have the following comments for the authors.

1. Clarification is needed to improve the methods.

- The specific questions to be addressed should be stated, the methods should be centered on those questions, and the inferences made from those questions should be clear. At the moment the paper reads like a 'large set of data, with some figures about the data' rather than as inference with a meaningful scope of inference.

- How many plants of each location/variety were used? What plant tissues? How many tissue samples? Without this information there can be no meaningful comparison among rose varieties / locations. Because there seems to be no replication there are no statistics regarding the comparisons of richness/etc. There should be no PCA without replication. There should be no bacterial community comparisons without replication.

- Please provide evidence that the sterilization method for the tissue surfaces remove the remnant DNA of epiphytes, even if the cells are not viable.

- Please provide the needed information about negative controls for the high-throughput sequencing.

- I don't understand the methods -- could the authors please clarify: it seems they cultured, then used high-throughput sequencing on the cultures?

- Far more information is needed about quality control and data preparation.

- A Venn diagram is a visualization, not an analysis per se; the authors should please revise the entire methods section to distinguish between visualization methods and actual analyses.

- The authors make statements that imply they have information about abundance; what positive controls were used to relate read abundance to biological abundance?

- I don't understand how the authors made functional gene predictions from 16S in a meaningful way. I understand the role of PIECRUST but would argue that far more information is needed to 100% validate these functional gene predictions. I strongly suggest much more caution in the presentation and interpretation of these results. We know of many bacteria with identical 16S but totally different functional traits.

2. The results, discussion, and conclusions generally are overlapping so much in content as to feel repetitive.

3. The authors state that all data are in the manuscript/supplements, but in the version I received the only supplement is an English-language editor's tracked changes.

6. PLOS authors have the option to publish the peer review history of their article (what does this mean?). If published, this will include your full peer review and any attached files.

Reviewer #1: Yes: Dr. Charu Lata

Reviewer #2: No

---

## [Author Response · Author response to Decision Letter 0]

15 Jan 2020

Reviewer #1: 

1. The criterion for selecting the 18 rose varieties is that these 18 varieties have a large planting area and a wide planting area in China. Each rose is planted over 10hm2 at the sampling site.

2. These sequences have not been submitted to GenBank and can be submitted later if necessary.

3. High throughput sequencing (HTS) technologies, such as 454 pyrosequencing and Illumina sequencing, have been extensively developed and used to explore the microbial diversity of Triticum aestivum L, Oryza sativa L, fue-cured tobaccos, Jasminum sambac (L.) Ait, etc.

4. The gene expression analysis results were obtained by comparing the functional analysis of metagenomic sequencing data with the corresponding 16s predictive function analysis results.

5. I have added more details to comment on the future prospects of the present study in the manuscript.

In all, I found the reviewer’s comments are quite helpful, and I revised my paper point-by-point. Thank you and the review again for your help! 

Reviewer #2: 

1.I quite appreciate your favorite consideration and insightful comments and found these comments are very helpful. I have modified it according to your suggestion.

The 18 rose varieties have a large planting area and a wide planting area in China. Each rose is planted over 10hm2 at the sampling site. Plant tissue is rose petals. Each rose petal sample comes from more than 3 roses, and the growth distance exceeds 100 meters. Sample processing reference Wang et al. (2018), (Wu et al., 2016) with some modifications. 

The surface of the rose sample is sterilized by rinsing with sterile water, and the last rinsed water (0.01mL) were spread onto the Plate Count Agar (PCA) to check the sterilization effect. Samples with completely sterile surfaces were rinsed with distilled water more than three times to remove the remaining microbial DNA.

Rose petal samples have negative controls when amplifying bacterial DNA, and only the successfully amplified samples are sequenced.

Because the bacterial DNA content directly amplified is relatively low in rose petal samples, Because of the low content of bacterial DNA directly amplified in rose petal samples, the microbial diversity was analyzed by liquid culture and enrichment. 

I describe more detail the quality control and data preparation information in the manuscript.

Regarding the venn diagram, I have modified it in the manuscript.

The functional gene predictions results were obtained by comparing the functional analysis of metagenomic sequencing data with the corresponding 16s predictive function analysis results. The accuracy of this method is 84% to 95%. The gene prediction is an inference in this article. The accuracy of 16S functional gene prediction can be referred to Langille et al. (2013). Thanks for your suggestion, I will more caution in the presentation and interpretation of these results.

2. I have revised the content of the results, discussion and conclusion.

3. If you need any more data, please don’t hesitate to contact me.

We appreciate for Reviewers’ warm work earnestly, and hope that the correction will meet with approval.

Once again, thank you very much for your comments and suggestions.

---

## [Decision Letter · Decision Letter 1]

4 Feb 2020

PONE-D-19-30114R1

Assessment of endophytic bacterial diversity in Rose by high-throughput sequencing analysis

PLOS ONE

Dear Liu_Yun-Guo,

Thank you for submitting your manuscript to PLOS ONE. After careful consideration, we feel that it has merit but does not fully meet PLOS ONE’s publication criteria as it currently stands. Therefore, we invite you to submit a revised version of the manuscript that addresses the points raised during the review process.

We would appreciate receiving your revised manuscript by Mar 20 2020 11:59PM. To enhance the reproducibility of your results, we recommend that if applicable you deposit your laboratory protocols in protocols.io, where a protocol can be assigned its own identifier (DOI) such that it can be cited independently in the future. For instructions see: http://journals.plos.org/plosone/s/submission-guidelines#loc-laboratory-protocols

We look forward to receiving your revised manuscript.

Kind regards,

Aradhana Mishra, Ph.D.

Academic Editor

PLOS ONE

Reviewers' comments:

Reviewer's Responses to Questions

**Comments to the Author**

1. If the authors have adequately addressed your comments raised in a previous round of review and you feel that this manuscript is now acceptable for publication, you may indicate that here to bypass the “Comments to the Author” section, enter your conflict of interest statement in the “Confidential to Editor” section, and submit your "Accept" recommendation.

Reviewer #1: All comments have been addressed

Reviewer #3: All comments have been addressed

2. Is the manuscript technically sound, and do the data support the conclusions?

Reviewer #1: Yes

Reviewer #3: Yes

3. Has the statistical analysis been performed appropriately and rigorously? 

Reviewer #1: Yes

Reviewer #3: Yes

4. Have the authors made all data underlying the findings in their manuscript fully available?

Reviewer #1: Yes

Reviewer #3: Yes

5. Is the manuscript presented in an intelligible fashion and written in standard English?

Reviewer #1: Yes

Reviewer #3: Yes

6. Review Comments to the Author

Reviewer #1: (No Response)

Reviewer #3: Comment

Although MS written well and illustrate the diversity; community composition of endophytes in native roses. The manuscript could be accepable for publication after the correction of following points.

1. MS should be check for plagiarism through authentic resources like turnitin or other acceptable software. This MS showed 22% plagiarism from different published resources like

i. Afusat Yinka Aregbe, Taihua Mu, Hongnan Sun. "Effect of different pretreatment on the microbial diversity of fermented potato revealed by high throughput sequencing" , Food Chemistry, 2019 (3% plagiarism from this paper).

ii. Amb-express.springeropen.com (3% plagiarism from this paper).

iii. Xinhui Wang, Songhu Wang, Hai Zhao. "Unraveling microbial community diversity and succession of Chinese Sichuan sausages during spontaneous fermentation by high throughput sequencing" , Journal of Food Science and Technology, 2019 (3% plagiarism from this paper).

Manuscript should be thoroughly revised. MS should not have match the same paragraph with other published article. Moreover, references mention in MS should be of original research articles.

2. References should be in journal format.

7. PLOS authors have the option to publish the peer review history of their article (what does this mean?). If published, this will include your full peer review and any attached files.

Reviewer #1: Yes: Dr. Charu Lata

Reviewer #3: Yes: sumit kumar soni

---

## [Author Response · Author response to Decision Letter 1]

10 Feb 2020

Reviewer #3: 

1. I have modified the content that may involve plagiarism.

2. The reference format has been modified.

We appreciate for Reviewers’ warm work earnestly, and hope that the correction will meet with approval.

---

## [Decision Letter · Decision Letter 2]

12 Mar 2020

Assessment of endophytic bacterial diversity in Rose by high-throughput sequencing analysis

PONE-D-19-30114R2

Dear Dr. Liu Yun-Guo,

We are pleased to inform you that your manuscript has been judged scientifically suitable for publication and will be formally accepted for publication once it complies with all outstanding technical requirements.

With kind regards,

Aradhana Mishra, Ph.D.

Academic Editor

PLOS ONE

Additional Editor Comments (optional):

Reviewers' comments:

Reviewer's Responses to Questions

**Comments to the Author**

1. If the authors have adequately addressed your comments raised in a previous round of review and you feel that this manuscript is now acceptable for publication, you may indicate that here to bypass the “Comments to the Author” section, enter your conflict of interest statement in the “Confidential to Editor” section, and submit your "Accept" recommendation.

Reviewer #1: All comments have been addressed

Reviewer #3: All comments have been addressed

2. Is the manuscript technically sound, and do the data support the conclusions?

Reviewer #1: Yes

Reviewer #3: Yes

3. Has the statistical analysis been performed appropriately and rigorously? 

Reviewer #1: Yes

Reviewer #3: Yes

4. Have the authors made all data underlying the findings in their manuscript fully available?

Reviewer #1: Yes

Reviewer #3: Yes

5. Is the manuscript presented in an intelligible fashion and written in standard English?

Reviewer #1: Yes

Reviewer #3: Yes

6. Review Comments to the Author

Reviewer #1: (No Response)

Reviewer #3: The comment raised in revision first has been corrected by the authors. The Manuscript now can be accepted in current form.

7. PLOS authors have the option to publish the peer review history of their article (what does this mean?). If published, this will include your full peer review and any attached files.

Reviewer #1: Yes: Charu Lata

Reviewer #3: Yes: Dr. Sumit Kumar Soni

---

## [Editor Report · Acceptance letter]

16 Mar 2020

PONE-D-19-30114R2 

Assessment of endophytic bacterial diversity in Rose by high-throughput sequencing analysis 

Dear Dr. Liu:

I am pleased to inform you that your manuscript has been deemed suitable for publication in PLOS ONE. Congratulations! Your manuscript is now with our production department. 

With kind regards,

on behalf of

Dr. Aradhana Mishra 

Academic Editor

PLOS ONE